

# The predictive role of preoperative serum glutamate dehydrogenase levels in microvascular invasion and hepatocellular carcinoma prognosis following liver transplantation—a single center retrospective study

Jinlong Gong[1,*], Yaxiong Li[2,*], Jia Yu[1], Tielong Wang[1], Jinliang Duan[1], Anbin Hu[1], Xiaoshun He[1] and Xiaofeng Zhu[1]

[1] Department of Organ Transplantation, The First Affiliated Hospital, Sun Yat-Sen University, Guangzhou, Guangdong province, China
[2] Department of Pancreato-Biliary Surgery, The First Affiliated Hospital, Sun Yat-Sen University, Guangzhou, Guangdong province, China
* These authors contributed equally to this work.

Corresponding author
Xiaofeng Zhu,
zhuxiaof@mail.sysu.edu.cn

## ABSTRACT

**Background:** As a critical metabolic substrate, glutamine is not only involved in the progression of many cancers but is also related to angiogenesis. Glutamate dehydrogenase (GLDH), a key enzyme in glutamine metabolism, has been reported to regulate tumor proliferation; however, its relationship with microvascular invasion (MVI) is unclear. This study evaluated the ability of preoperative serum GLDH levels to predict MVI and the long-term survival of hepatocellular carcinoma (HCC) patients after liver transplantation (LT).

**Methods:** HCC patients that underwent LT from January 2015 to May 2020 at the First Affiliated Hospital of Sun Yat-Sen University were enrolled in our retrospective analysis. Clinicopathological variables were extracted from medical records. A receiver operating characteristic curve was created to determine the optimal cut-off value of GLDH for MVI.

**Results:** Preoperative GLDH was significantly elevated in the MVI-positive group ($U = 454.00$, $p = 0.000$). The optimal cut-off value of GLDH for MVI was 7.45 U/L, with an area under the curve of 0.747 (95% CI [0.639–0.856], $p = 0.000$). The sensitivity was 79.3%, while the specificity was 64.5%. GLDH > 7.45 U/L ($p = 0.023$) and maximum diameter >5 cm ($p = 0.001$) were independent risk factors for the presence of MVI. Patients with GLDH > 7.45 U/L had significantly poorer overall survival ($p = 0.001$) and recurrence-free survival ($p = 0.001$) after LT than patients with GLDH ≤ 7.45 U/L. Similarly, patients with MVI were associated with poor survival ($p = 0.000$).

**Conclusions:** Preoperative elevated serum GLDH levels predict MVI and poorer long-term survival for HCC after LT.

## INTRODUCTION

Hepatocellular carcinoma (HCC) accounts for 85–90% of all primary liver cancer cases (*Zhou et al., 2018*). It is the fourth most common malignancy and the third leading cause of tumor-related deaths in China (*Chen et al., 2016*). Liver transplantation (LT) is considered the definitive treatment option for HCC as it removes not only the tumor but also the cirrhotic liver, which has the potential to develop new HCC lesions (*Yang et al., 2019a*). However, recurrence post-LT is a major problem that threatens the long-term survival of HCC patients. Currently, even if the Milan or University of California San Francisco (UCSF) criteria are used to select transplant recipients, the probability of HCC recurrence post-LT ranges from 16% to 33% (*Verna et al., 2020*; *Xu et al., 2016*), with the median survival post-recurrence being only 10.6–12.2 months (*Bodzin et al., 2017*; *Sapisochin et al., 2015*). These numbers indicate that the traditional recipient selection and organ allocation criteria, which are based on tumor burden, do not always reflect tumor biology. Alternatively, molecular biomarkers may be superior in revealing tumor aggressiveness (*Felden & Villanueva, 2020*).

Given the scarcity of available organs for transplantation, the recurrence risk in HCC patients should be evaluated preoperatively to improve recipient selection strategy and to develop individualized immunosuppressive and anti-cancer regiments. Microvascular invasion (MVI) is widely recognized as a powerful adverse predictor for HCC recurrence after LT (*Victor et al., 2020*; *Mehta et al., 2017*). However, MVI is usually difficult to assay before surgery because it depends on pathological examination. Therefore, the development of a noninvasive method that can accurately identify MVI preoperatively has become a research hotspot (*Xu et al., 2019*; *Ma et al., 2020*).

Glutamate dehydrogenase (GLDH) belongs to the amino acid dehydrogenase enzyme superfamily. It catalyzes the reversible inter-conversion of glutamate to α-ketoglutarate and ammonia using NADP(H) and/or NAD(H) as cofactors and plays an important role in nitrogen and carbon metabolism (*Oliveira et al., 2012*; *Spanaki, Kotzamani & Plaitakis, 2017*). Glutamine is an important source of metabolic energy in many cancers (*Matés et al., 2019*). Consequently, as the main enzyme regulating glutamate metabolism, GLDH may play a crucial role in tumor growth (*Spinelli et al., 2017*; *Jin et al., 2015*). *Jin et al. (2018)* found that GLDH1 mediates the metabolic reprogramming of glutaminolysis and regulates lung cancer metastasis, subsequently *Yang et al. (2020)* demonstrated that GLDH1-mediated glutaminolysis is associated with epidermal growth factor-promoted glioblastoma cell proliferation. Moreover, a clinical study (*Piras-Straub et al., 2015*) reported that GLDH is an independent predictor for HCC recurrence post-LT.

Glutamine metabolism has also been reported to be essential in angiogenesis (*Huang et al., 2017*). Consequently, we hypothesize that, given its role in glutamate metabolism, GLDH can serve as a serum biomarker for MVI. To date, no studies have examined

the relationship between GLDH and MVI in HCC. Accordingly, we evaluated the ability of preoperative serum GLDH levels in predicting MVI and the survival of HCC patients following LT.

## PATIENTS AND METHODS

### Ethical statement

This study was approved by the Institutional Ethics Committee for Clinical Research and Animal Trials of the First Affiliated Hospital of Sun Yat-sen University (Ethical Application Ref:2021-352). The ethical principles were in line with the Declaration of Helsinki. All patients signed an informed consent form before LT. The sole source of LTs in our study were from civilian liver donations; no organs from executed prisoners were transplanted.

### Patient selection

Patients who underwent LT at the First Affiliated Hospital of Sun Yat-Sen University from January 2015 to May 2020 were recruited for our study. The inclusion criteria were: underwent LT and pathological confirmation of HCC. The exclusion criteria were: patients with a history of hepatectomy, transarterial chemoembolization, or ablation before LT; cases with incomplete medical records that made it impossible to extract all needed clinicopathological parameters; perioperative death in hospital. Distant metastases or other concurrent malignancies were considered contraindications for LT.

### Surgical technique and immunosuppression regimens

Generally, classic orthotopic LT was performed for the HCC patients. In patients who had renal insufficiency before surgery, piggyback LT was chosen cautiously. Venovenous bypass was not used. Our institution's immunosuppression regimen consisted of two doses of Basiliximab (20 mg each) during LT and on the fourth day post-LT. Inhibitors of calcineurin (usually Tacrolimus, in a few cases cyclosporine) and mechanistic target of rapamycin (mTOR) (usually Sirolimus) were used for maintenance immunosuppression. Depending on the side effects, the specific maintenance immunosuppression regimen was adjusted during the follow-up.

### Patient variable extraction and follow-up

Patient variables were extracted from the electronic medical records. All serum biochemical variables were taken from the results of the last preoperative test. Tumor-related variables, including tumor size, number, and liver cirrhosis, were based on the results of the last preoperative radiological examination. MVI was diagnosed post-LT *via* pathological examination.

After being discharged, patients were routinely followed-up as outpatients or with phone calls. Elevated serum alpha-fetoprotein (AFP) was considered as a sign of recurrence; however, recurrence was defined based on radiological findings of recurrent lesions using contrast-enhanced CT/MRI or ultrasound. Overall survival (OS) was

calculated from the date of LT to the date of death or last follow-up, while recurrence-free survival (RFS) was calculated from the date of LT to the date of recurrence.

## Statistical analysis

Statistical analyses were carried out using SPSS (version 23.0, IBM). Data that were not normally distributed are presented as median value and the $M$ (range), and differences between groups were detected *via* the Mann-Whitney $U$ test. The optimal cut-off value of GLDH was determined using receiver operating characteristic (ROC) curve analysis. To identify the risk factors and independent risk factors for MVI, univariate and multivariate logistic regression analyses were performed. Baseline variables that showed univariate significance were entered into the multivariate logistic analysis. OS and RFS were calculated using the Kaplan-Meier method and compared using the log-rank test. In all analyses, $p < 0.05$ was considered statistically significant.

## RESULTS

### Patient characteristics

In total, 91 patients were ultimately enrolled in our study. A total of 92.31% (84) were male, while only seven (7.69%) were female. Most (67, 73.63%) patients were under 60 years old, while 24 (26.37%) were over 60. Sixty-two (68.13%) patients had AFP levels under 200 μg/L; the remaining 29 (31.87%) had elevated AFP levels (*i.e.*, greater than 200 μg/L). The majority (90.11%) of patients were hepatitis B surface antigen-positive; accordingly, nearly 90% (84.62%) of the patients had hepatitis B virus-related liver cirrhosis. The radiological examinations revealed that 42 patients (46.15%) had multiple tumors, and 43 (47.25%) had maximum tumor diameters greater than 5 cm. MVI was confirmed *via* postoperative pathological examination in 29 (31.87%) patients.

The following variables did not follow a normal distribution that are expressed as median value and $M$ (range): albumin/globulin: 1.20 (0.50–3), prealbumin (PA): 95 (38–365) mg/L, total bilirubin (TB): 37.30 (9–584) μmol/L, alanine aminotransferase (ALT): 30 (4–825) U/L, cancer antigen 125 (CA125): 61.90 (4.90–4854.80) U/mL, and GLDH: 7.30 (1.30–918.80) U/L (Table 1).

### Expression profiles and the optimal cut-off value of GLDH

As illustrated in Fig. 1A, preoperative serum GLDH levels were significantly higher in the MVI-positive patients ($U = 454.00$, $p = 0.000$).

Next, we conducted a ROC curve analysis to determine the optimal cut-off value of GLDH for the presence of MVI (Fig. 1B). The optimal cut-off value of GLDH was 7.45 U/L for MVI, and the area under the curve was 0.747 (95% CI [0.639–0.856], $p = 0.000$). The sensitivity and specificity were, respectively, 79.3% and 64.5%. The maximum Youden index was 0.438.

### The clinical value of GLDH in predicting MVI

GLDH levels were divided into categorical variables (>7.45 U/L and ≤7.45 U/L) based on the optimal cut-off value from the ROC curve. The univariate logistic regression analysis

**Table 1 Baseline characteristics of 91 patients.**

| Variables | No. of patients (%)/Median (M) |
|---|---|
| Sex | |
| Male | 84 (92.31) |
| Female | 7 (7.69) |
| Age (years) | |
| ≤60 | 67 (73.63) |
| >60 | 24 (26.37) |
| AFP (µg/L) | |
| ≤200 | 62 (68.13) |
| >200 | 29 (31.87) |
| HBsAg | |
| Negative | 9 (9.89) |
| Positive | 82 (90.11) |
| Maximum diameter (cm) | |
| ≤5 | 48 (52.75) |
| >5 | 43 (47.25) |
| Tumor number | |
| Single | 49 (53.85) |
| Multiple | 42 (46.15) |
| MVI | |
| Negative | 62 (68.13) |
| Positive | 29 (31.87) |
| Liver cirrhosis | |
| Negative | 14 (15.38) |
| Positive | 77 (84.62) |
| Albumin/Globulin | 1.20 (0.50–3) |
| PA (mg/L) | 95 (38–365) |
| TB (µmol/L) | 37.30 (9–584) |
| ALT (U/L) | 30 (4–825) |
| CA125 (U/mL) | 61.90 (4.90–4854.80) |
| GLDH (U/L) | 7.30 (1.30–918.80) |

Note:
AFP, alpha-fetoprotein; HBsAg, hepatitis B surface antigen; MVI, microvascular invasion; PA, prealbumin; ALT, alanine aminotransferase; GLDH, glutamate dehydrogenase.

indicated that AFP > 200 µg/L ($p = 0.000$), maximum diameter >5 cm ($p = 0.000$), liver cirrhosis ($p = 0.008$), and GLDH > 7.45 U/L ($p = 0.000$) were risk factors for MVI in HCC. Furthermore, the multivariate analysis revealed that maximum diameter >5 cm ($p = 0.001$) and GLDH > 7.45 U/L ($p = 0.023$) were independent risk factors for MVI (Table 2).

## Prognostic significance of GLDH in the OS and RFS of HCC patients

We produced Kaplan-Meier curves to evaluate the effect of high preoperative serum GLDH levels on the long-term survival of HCC patients following LT. As depicted in
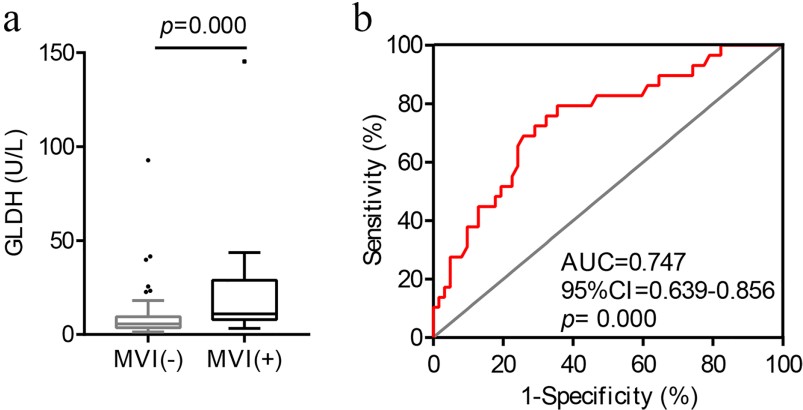

**Figure 1 The correlation and optimal cut-off value between preoperative serum GLDH levels and MVI.** (A) Distribution of GLDH levels in patients with and without MVI. (B) ROC curve of GLDH for MVI. AUC, area under the curve; GLDH, glutamate dehydrogenase; MVI, microvascular invasion; ROC, receiver operating characteristic.

**Table 2 Univariable and multivariable logistic analysis of risk factors to MVI**

| Variables | OR | Univariate analysis | | OR | Multivariate analysis | |
|---|---|---|---|---|---|---|
| | | HR (95% CI) | *p* | | HR (95% CI) | *p* |
| Sex (male *vs* female) | | | 0.999 | | | |
| Age, y (>60 *vs* ≤60) | 0.64 | [0.22–1.83] | 0.402 | | | |
| AFP, μg/L (>200 *vs* ≤200) | 5.90 | [2.24–15.59] | 0.000 | 3.06 | 0.96–9.75 | 0.059 |
| HBsAg (positive *vs* negative) | 1.72 | [0.33–8.84] | 0.517 | | | |
| Maximum diameter, cm (>5 *vs* ≤5) | 15.28 | [4.65–50.19] | 0.000 | 8.85 | 2.41–32.56 | 0.001 |
| Tumor number (multiple *vs* single) | 1.39 | [0.57–3.36] | 0.467 | | | |
| Liver cirrhosis (positive *vs* negative) | 0.20 | [0.06–0.65] | 0.008 | 0.75 | 0.19–3.02 | 0.687 |
| Albumin/Globulin | 0.52 | [0.17–1.65] | 0.267 | | | |
| PA (mg/L) | 1.01 | [1.00–1.01] | 0.185 | | | |
| TB (μmol/L) | 0.99 | [0.98–1.00] | 0.122 | | | |
| ALT (U/L) | 1.00 | [1.00–1.01] | 0.248 | | | |
| CA125 (U/mL) | 1.00 | [1.00–1.00] | 0.343 | | | |
| GLDH, U/L (>7.45 *vs* ≤7.45) | 6.97 | [2.47–19.68] | 0.000 | 4.01 | 1.21–13.34 | 0.023 |

**Note:**
AFP, alpha-fetoprotein; HBsAg, hepatitis B surface antigen; MVI, microvascular invasion; PA, prealbumin; ALT, alanine aminotransferase; GLDH, glutamate dehydrogenase.

Figs. 2A and 2B, patients with GLDH > 7.45 U/L had significantly poorer OS (*p* = 0.001) and RFS (*p* = 0.001) than patients with GLDH ≤ 7.45U/L. Similarly, as a powerful predictor, the predictive ability of MVI has also been verified in our cohort. Those patients with MVI were associated with worse survival (Figs. 2C, 2D).

## DISCUSSION

This study found that elevated preoperative serum GLDH levels is an effective predictor for MVI in HCC and is associated with poor prognosis post-LT. The ROC curve indicated that the optimal cut-off value of GLDH for MVI was 7.45 U/L, with a sensitivity and

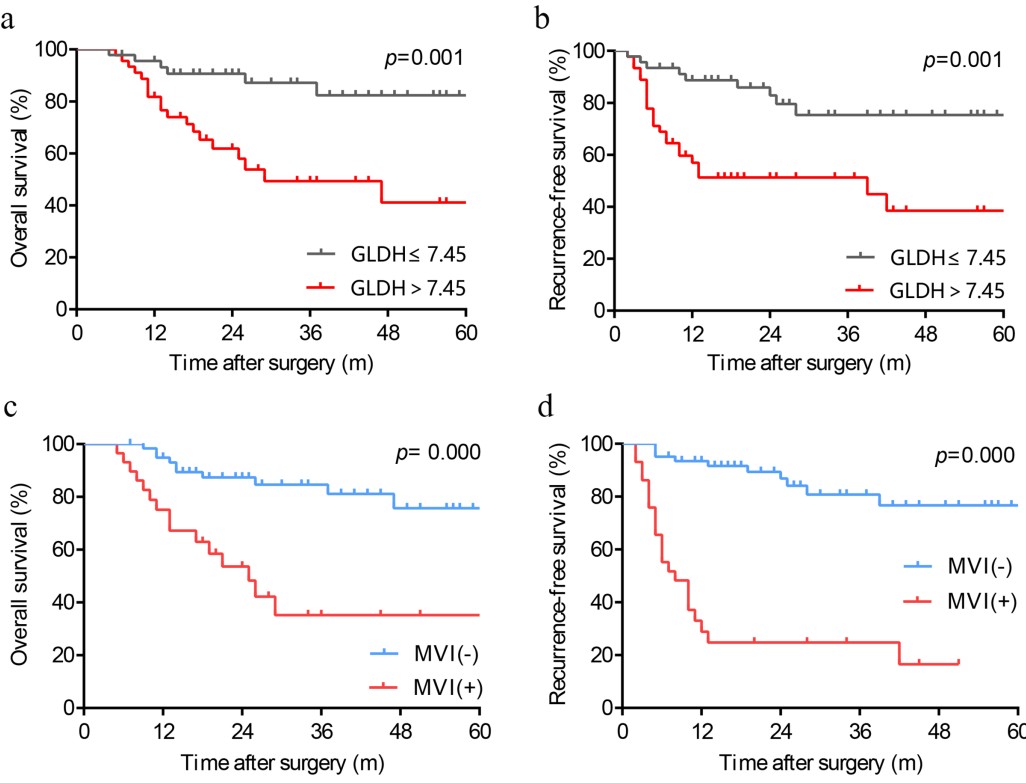

**Figure 2** Kaplan-Meier analysis in patients with different levels of GLDH (A, B) and patients with or without MVI (C, D). GLDH, glutamate dehydrogenase; MVI, microvascular invasion.

specificity of 79.3% and 64.5%, respectively. The multivariate analysis found that GLDH > 7.45 U/L and maximum diameter >5 cm were independent risk factors for MVI. Moreover, Kaplan-Meier curves revealed that higher preoperative levels of GLDH was associated with significantly poorer OS and RFS post-LT in HCC patients.

Recurrence has become the greatest adverse factor in decreasing the survival of HCC patients after LT. Thus, effectively predicting the risk of tumor recurrence before surgery so that corresponding treatments can be carried out is vital. Since MVI is widely recognized as a powerful risk factor for HCC recurrence, the prediction of MVI may be used as a proxy of recurrence risk. Previous studies (*Lee et al., 2017*; *Yang et al., 2019b*) have reported some progress in predicting MVI based on radiological evidence. Additionally, serum biomarkers such as the neutrophil-to-lymphocyte ratio or circulating tumor cells have demonstrated some promise in this area (*Nitta et al., 2019*; *Zhou et al., 2020*).

Our results demonstrate the potential value of GLDH in predicting MVI from the clinical perspective. We hypothesize that this predictive ability may be due to its critical role in glutamine metabolism. GLDH catalyzes both the oxidative deamination of glutamate to α-ketoglutarate as well as the reductive amination in the reverse direction

(*Hohnholt et al., 2018*). In addition to being an energy substrate for tumor metabolism, glutamine is essential in angiogenesis. *Kim et al. (2017)* found that glutamine serves as a nitrogen source in the generation of biomass in endothelial cell proliferation. Furthermore, *Huang et al. (2017)* reported that glutamine metabolism is essential for vessel sprouting *in vitro* and *in vivo*. Therefore, as a key enzyme in glutamine metabolism, GLDH likely regulates angiogenesis. Nevertheless, to the best of our knowledge, no studies have, to date, directly confirmed the relationship between GLDH and tumor-angiogenesis-related physiological processes. Basic research may be needed to elucidate the potential mechanisms underlying the regulation of angiogenesis by GLDH. As well, our results confirmed the effectiveness of maximum tumor diameter in predicting MVI, which is consistent with a previous study (*Yan et al., 2020*).

The cut-off value of GLDH predicting MVI in our study was found to be 7.45 U/L, very close to the upper limit of the normal level (7.5 U/L) in our center. The possible reasons are listed as followings: Firstly, for a specific indicator, the reference ranges of the normal level in different centers are common to be different, mainly owing to different detecting methods. Secondly, the clinical significance of our study lies in indicating the existence of MVI in HCC through the increasing trend in GLDH. In contrast, no consensus exists on the specific cut-off value of GLDH, mainly due to the small sample size of our study. Therefore, studies with large samples are necessary to explore and verify the optimal cut-off value of the GLDH in future. In addition, the population characteristics of patients in each region are also different, which may introduce some individual differences into the statistical analysis.

Our results may be effective in stratifying recurrence risk in HCC patients pre-LT and may provide a reference in designing immunosuppression and anti-tumor regimens. Patients with significantly elevated preoperative serum GLDH levels may need closer monitoring after LT.

The major limitation of this study was the small sample size as mentioned above. To ensure the accuracy of the pathological diagnoses, only cases that did not undergo anti-tumor therapy before LT were enrolled in our study. Transarterial chemoembolization/ablation may result in the necrosis of HCC lesions, which creates difficulties in identifying MVI under the microscope. Thus, larger sample sizes will be needed in future studies.

In conclusion, our study identified the potential clinical value of GLDH in predicting MVI and the long-term prognosis of HCC patients after LT.

### Funding
The authors received no funding for this work.

### Competing Interests
The authors declare that they have no competing interests.

## Author Contributions

- Jinlong Gong conceived and designed the experiments, performed the experiments, analyzed the data, authored or reviewed drafts of the paper, and approved the final draft.
- Yaxiong Li performed the experiments, analyzed the data, prepared figures and/or tables, and approved the final draft.
- Jia Yu performed the experiments, prepared figures and/or tables, and approved the final draft.
- Tielong Wang performed the experiments, prepared figures and/or tables, and approved the final draft.
- Jinliang Duan performed the experiments, prepared figures and/or tables, and approved the final draft.
- Anbin Hu conceived and designed the experiments, authored or reviewed drafts of the paper, and approved the final draft.
- Xiaoshun He conceived and designed the experiments, authored or reviewed drafts of the paper, and approved the final draft.
- Xiaofeng Zhu conceived and designed the experiments, authored or reviewed drafts of the paper, and approved the final draft.

## Human Ethics

The following information was supplied relating to ethical approvals (*i.e.*, approving body and any reference numbers):

The study was approved by the Institutional Ethics Committee for Clinical Research and Animal Trials of the First Affiliated Hospital of Sun Yat-sen University (Ethical Application Ref:2021-352).

## Data Availability

The raw data is provided in the File S1. All statistical analysis data were derived from the raw data.

## Supplemental Information

Supplemental information for this article can be found online at http://dx.doi.org/10.7717/peerj.12420#supplemental-information.

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
