# Peer review of "The predictive role of preoperative serum glutamate dehydrogenase levels in microvascular invasion and hepatocellular carcinoma prognosis following liver transplantation—a single center retrospective study"

_PeerJ, doi:10.7717/peerj.12420_

## Round 0.1 · original submission · Major Revisions

The reviewers found the manuscript interesting and important, however, before it can be considered for publication, it needs some improvements. When revising the manuscript, all issues raised by both reviewers should be addressed, in particular Reviewer 2's concerns regarding the cut-off point for GLDH and the mechanism of increasing GLDH in patients. The manuscript must be edited by a professional English editor before resubmitting.

Reviewer 1 ·

Basic reporting

1. The manuscript would benefit from editing to correct English grammar.

2. In Table 1, please provide mean or median values for the lab test results, in addition to the ranges.

3. In Figure 1A, please provide a box plot or dot plot instead of a bar graph. Box and dot plots provide far more information about the distribution of the results, which will aid interpretation by the reader.

Experimental design

1. In Figure 2, the authors should compare GLDH with MVI to predict survival.

Validity of the findings

No comment.

Additional comments

No additional comments.

Reviewer 2 ·

Basic reporting

no comment

Experimental design

no comment

Validity of the findings

This is a well written manuscript that addresses a relevant subject matter. The ability for GLDH to predict MVI and poor long-term survival for HCC after liver transplant could profoundly effect the selection of transplant recipients.

However, the normal reference range for GLDH is between 2 U/L and 10 U/L with an upper limit of normal of 11 U/L. As the cut-point for GLDH to predict MVI and poor long-term survival for HCC after liver transplant is 7.45 U/L. It is unclear to this reviewer how GLDH is able to discriminate outcomes in this normal range. Furthermore, GLDH is released following hepatocyte injury. How do subtle changes in GLDH caused by drug-induced liver injury or other hepatotoxic events in these patients affect the interpretation? My primary concern is that the cut-point for GLDH is in normal range for the biomarker.

It may be worthwhile to evaluate the correlation between GLDH and ALT. This may allow an understanding for the role of hepatocellular injury and an increase in GLDH in these patients. We have previously evaluated both ALT and GLDH in patients post transplant and have observed a disconnect between these serum biomarkers. It is not clear if this observation is linked to the authors observation.

Additional comments

no comment

---

## Round 0.2 · Minor Revisions

While you adequately responded to Reviewer 2's concerns in your rebuttal letter, one point was not fully covered in the manuscript. The latter concerns the cut-off point for GLDH, which for Reviewer 2 appears to be within the normal range for the biomarker. To make this point clear to a wider group of interested readers from different centers, please discuss the possible causes of the cut-off values in more detail in the Discussion section. Please make sure the English in the paragraph you added is edited.

Reviewer 1 ·

Basic reporting

Reporting is acceptable.

Experimental design

Design is acceptable.

Validity of the findings

Findings are acceptable as reported.

Additional comments

No additional comments.

---

## Round 0.3 · accepted · Accept

The introduced corrections improved the manuscript, which made it suitable for publication.